# RAR Inhibitors Display Photo-Protective and Anti-Inflammatory Effects in A2E Stimulated RPE Cells In Vitro through Non-Specific Modulation of PPAR or RXR Transactivation

**DOI:** 10.3390/ijms25053037

**Published:** 2024-03-06

**Authors:** Valérie Fontaine, Thinhinane Boumedine, Elodie Monteiro, Mylène Fournié, Gendre Gersende, José-Alain Sahel, Serge Picaud, Stanislas Veillet, René Lafont, Mathilde Latil, Pierre J. Dilda, Serge Camelo

**Affiliations:** 1Sorbonne Université, INSERM, CNRS, Institut de la Vision, 17 Rue Moreau, 75012 Paris, France; valerie.fontaine@inserm.fr (V.F.); thinhinane.boumedine@inserm.fr (T.B.); mylene.fournie@gmail.com (M.F.); sahelja@upmc.edu (J.-A.S.); serge.picaud@inserm.fr (S.P.); 2Fondation Ophtalmologique Rothschild, 29 rue Manin, 75019 Paris, France; 3Department of Ophthalmology, The University of Pittsburgh School of Medicine, Pittsburgh, PA 15213, USA; 4Biophytis, Sorbonne Université, BC9, 4 place Jussieu, 75005 Paris, Francemathilde.latil@biophytis.com (M.L.); pierre.dilda@biophytis.com (P.J.D.)

**Keywords:** *N*-retinylidene-*N*-retinylethanolamine (A2E), angiogenesis, inflammation, norbixin, nuclear receptor (NR), peroxisome proliferator-activated receptor (PPAR), retinoic acid receptor (RAR), retinal pigment epithelium (RPE), retinoic X receptor (RXR)

## Abstract

*N*-retinylidene-*N*-retinylethanolamine (A2E) has been associated with age-related macular degeneration (AMD) physiopathology by inducing cell death, angiogenesis and inflammation in retinal pigmented epithelial (RPE) cells. It was previously thought that the A2E effects were solely mediated via the retinoic acid receptor (RAR)-α activation. However, this conclusion was based on experiments using the RAR “specific” antagonist RO-41-5253, which was found to also be a ligand and partial agonist of the peroxisome proliferator-activated receptor (PPAR)-γ. Moreover, we previously reported that inhibiting PPAR and retinoid X receptor (RXR) transactivation with norbixin also modulated inflammation and angiogenesis in RPE cells challenged in the presence of A2E. Here, using several RAR inhibitors, we deciphered the respective roles of RAR, PPAR and RXR transactivations in an in vitro model of AMD. We showed that BMS 195614 (a selective RAR-α antagonist) displayed photoprotective properties against toxic blue light exposure in the presence of A2E. BMS 195614 also significantly reduced the AP-1 transactivation and mRNA expression of the inflammatory interleukin (IL)-6 and vascular endothelial growth factor (VEGF) induced by A2E in RPE cells in vitro, suggesting a major role of RAR in these processes. Surprisingly, however, we showed that (1) Norbixin increased the RAR transactivation and (2) AGN 193109 (a high affinity pan-RAR antagonist) and BMS 493 (a pan-RAR inverse agonist), which are photoprotective against toxic blue light exposure in the presence of A2E, also inhibited PPARs transactivation and RXR transactivation, respectively. Therefore, in our in vitro model of AMD, several commercialized RAR inhibitors appear to be non-specific, and we propose that the phototoxicity and expression of IL-6 and VEGF induced by A2E in RPE cells operates through the activation of PPAR or RXR rather than by RAR transactivation.

## 1. Introduction

Age-related macular degeneration (AMD) is the commonest cause of severe visual loss and blindness in developed countries among individuals aged 60 and older and is still a major medical need [1,2,3]. AMD slowly progresses from early AMD to intermediate AMD (iAMD), which further evolves to the late stages of the disease: neovascular AMD and/or geographic atrophy (GA) [3]. Neovascular AMD, also called “wet” AMD, results from choroidal neovascularization and hemorrhage in the sub-retinal space at the level of the macula that are mostly driven by vascular endothelial growth factor (VEGF) hypersecretion [4]. Meanwhile “dry” AMD is characterized by slowly expanding degenerative lesions of photoreceptors and the retinal pigmented epithelium (RPE), leading to the progressive retinal degeneration and dysfunction [5]. GA, which is the final stage of dry AMD, displays extensive retinal atrophy patches linked with severe visual impairment when present at the level of the macula [5].

Accumulation of *N*-retinylidene-*N*-retinylethanolamine (A2E), which is a by-product of the visual cycle [6] in the RPE [7] and Bruch’s membrane [8], was proposed to be associated with the initiation of the disease. For instance, in vitro, A2E in the presence of blue light illumination is toxic for RPE cells [9,10,11]. Moreover, A2E alone increases the secretion of inflammatory cytokines and VEGF by RPE cells in vitro and in AMD animal models in vivo [12,13,14,15].

The recently demonstrated interaction of A2E with several nuclear receptors (NRs) could explain these effects [16]. NRs are considered “metabolic sensors” that can bind multiple ligands. Once activated, NRs become transcription factors that lead to the activation or repression of specific target genes, thus modulating many biological pathways, including metabolism, cell death, inflammation and angiogenesis, which are all involved in AMD pathogenesis [16,17,18,19,20]. It was demonstrated that A2E binds to and induces the transactivation of the α-isoform of the retinoic acid receptor (RAR) [14]. RAR transactivation inhibition with RO-41-5253, which is an RAR-α antagonist [21], reduces A2E-induced VEGF production in vitro and in vivo [14,15] and RPE cell death in vivo [15]. This led to the logical conclusion that RAR is central in mediating these biological effects induced by A2E. However, it was shown that RO-41-5253 is not only an RAR-α antagonist but is also an agonist of peroxisome proliferator-activated receptor (PPAR)-γ [22]. Moreover, we previously reported that the inhibition of PPAR and retinoid X receptors (RXR) transactivation by 9-cis-norbixin (norbixin (NBX)), which is a di-apocarotenoid extracted from Bixa Orellana seeds, also modulates inflammation and angiogenesis in RPE cells stimulated by A2E. These observations suggest that these NRs, rather than RAR, might be involved [23]. Here, by using several RAR antagonists, we deciphered the respective roles of RAR, PPAR and RXR transactivations in cell death, angiogenesis and inflammation induced by A2E in this in vitro model of AMD. We confirmed that A2E induces RAR transactivation, and we showed that BMS 195614, which is a selective RAR-α antagonist [24], provides photoprotection and also reduces the expression of inflammatory interleukin (IL)-6 and VEGF enhanced by A2E. However, we report that in addition to their selective antagonism of RAR-α transactivation, BMS 195614, as well as other antagonists of RAR, which protect RPE cells against A2E stimulation in the presence of blue light, displayed nonspecific inhibition of PPAR or RXR transactivations. In conclusion, we propose that photo-protection and modulation of A2E’s biological effects provided by RAR inhibitors in vitro could be mediated via these non-specific modulations of PPAR or RXR transactivations rather than by their targeted inhibition of RAR transactivation.

## 2. Results

### 2.1. A2E Bound RAR-α and Induced the Transactivation of RAR

It was previously demonstrated that A2E binds to and induces the transactivation of the α-isoform of the retinoic acid receptor (RAR) [14]. First, we wished to confirm that this was indeed the case in our in vitro model using primary porcine RPE cells by evaluating the binding profile of A2E to the α-isoform of RAR. We performed competition experiments in vitro between A2E and all-*trans*-retinoic acid (ATRA), which is a pan-agonist of RARs and has a far greater affinity. The IC_50_ of ATRA was 9.5 10^−9^ M in our hands and was previously reported to be 0.49 10^−9^ M [14]. We determined that A2E bound to RAR-α with an IC_50_ of 2.8 × 10^−6^ M and a K_D_ of 1.4 × 10^−6^ M (Figure 1A). The control vehicle alone did not interfere with the binding of [^3^H] ATRA to RAR-α, confirming the specific binding of A2E. Then, the effects of A2E on the endogenous RAR transactivation in porcine RPE cells were investigated by means of a luciferase assay. The level of transactivation of RAR by A2E was determined in porcine RPE cells transfected with a plasmid-expressing luciferase under the control of the RAR response element. Dose–response experiments from 1 to 17.5 μM showed that the minimal required concentration of A2E to transactivate RAR in porcine RPE cells was 5 μM (Figure 1B). However, the transactivation of RAR by A2E was weak compared with the effect of ATRA since 17.5 μM of A2E was necessary to induce the transactivation of RAR at similar levels than a lower concentration of ATRA (0.1 μM) (Figure 1C).

### 2.2. The RAR Inhibitor BMS 195614 Reduced the Phototoxicity of RPE Cells Induced by Blue Light in the Presence of A2E

Previous studies using the RAR antagonist RO-41-5253 suggested the importance of RAR modulation in the toxicity of A2E in the presence of blue light for RPE cells [14,15]. However, RO-41-5253 also induces the non-specific activation of PPAR-γ, casting doubts on the validity of these previous observations [22]. To confirm the importance of RAR, here we tested the capacity of the specific synthetic RAR-α antagonist BMS 195614 to protect RPE cells against A2E-induced phototoxicity in vitro. At 10 μM, BMS 195614 was able to reduce the death of primary porcine RPE cells exposed to blue light in the presence of 30 μM of A2E (*p* < 0.001) (Figure 2). This effect was comparable with the photoprotection provided by norbixin at 20 µM (Figure 2). This result tends to confirm that the inhibition of RAR is important for limiting the phototoxicity induced by blue light in the presence of A2E.

### 2.3. The RAR-α Antagonist BMS 195614 and Norbixin Restored the Expression of Bcl2 Downregulated by A2E In Vitro

Since we previously showed that A2E binds to RAR-α, as well as the protective effect of the RAR-α antagonist BMS 195614 on the phototoxicity of blue light in the presence of A2E, we evaluated the levels of Bcl2, which is an anti-apoptotic protein known to protect RPE cells in vitro [25]. Here, we showed that the exposure of primary porcine RPE cells to 30 μM of A2E significantly reduced the expression of Bcl2 at the protein level (17.1% of control [DMSO], Figure 3A, *p* < 0.001). The RAR-α antagonist BMS 195614 at 10 μM was able to partially maintain the protein expression of Bcl2 (Figure 3A, 49.7% of the control [DMSO]), but this was not statistically significant (Figure 3C). Similarly, norbixin at 20 μM partially but significantly maintained the Bcl2 expression that was downregulated by A2E in the RPE cells in vitro (Figure 3B, 44.7% of the control [DMSO]). The quantification of Bcl2 inhibition by A2E and its restoration by NBX was statistically significant *p* < 0.01). The western blots of the effects of BMS 195614 and of NBX on Bcl2 expression are available in Appendix A respectively. Altogether, these observations suggest that RAR inhibition could play an important role in the regulation of Bcl2 expression, which could explain the photoprotective properties of norbixin and BMS 195614 on the RPE cells challenged by blue light in the presence of A2E.

### 2.4. The RAR-α Antagonist BMS 195614 Inhibited A2E Induction of IL-6 and VEGF mRNA Expression In Vitro

Then, we analyzed the effects of RAR inhibition on A2E-induced inflammation and angiogenesis in the RPE cells in vitro. First, we compared the transactivation of NF-κB and AP-1 induced by A2E in the presence or absence of BMS 195614. NF-κB transactivation induced by A2E was strongly downregulated by BMS 195614 (−51.75% vs. A2E alone, *p* < 0.05, Figure 4A). The transactivation of AP-1, which is another key modulator in inflammatory cytokine expression induced by A2E, was also significantly inhibited by BMS 195614 (−44.6% vs. A2E alone, *p* < 0.05, Figure 4B). Accordingly, BMS 195614 also significantly reduced the mRNA expression of IL-6 (−172% vs. A2E alone and −133% vs. control [DMSO], *p* < 0.001 for both comparisons, Figure 4C), and the mRNA expression of VEGF, which is the major driver of neovascular AMD induced by A2E, in the porcine RPE cells in vitro (Figure 4D, −72.8% vs. A2E alone, *p* < 0.01). This indicates that the RAR-α antagonist BMS 195614 modulated inflammation and angiogenesis induced by A2E in the RPE cells in vitro. Overall, our observations agree with the important role of RAR in mediating A2E’s biological effects.

### 2.5. Differential Effects of BMS 195614 and Norbixin in Nuclear Receptor Transactivations in RPE Cells In Vitro

So far, our observations support an important role of RAR in mediating A2E’s biological effects. This hypothesis was apparently further confirmed by the fact that the RAR-α antagonist BMS 195614 reduced the RAR transactivation induced by A2E by 147.8% (*p* < 0.001) (Figure 5A). However, we previously published that norbixin protects the primary porcine RPE cells against A2E-mediated phototoxicity and limits VEGF and inflammatory cytokines mRNA expression, and this was associated with the inhibition of the transactivation of PPAR and RXR induced by A2E [23,26,27]. To confirm the role of RAR in A2E’s biological effects, we tested the effect of norbixin on the RAR transactivation induced by A2E. Surprisingly, norbixin at 20 μM was not able to reduce the transactivation of RAR in RPE cultivated in the presence of A2E. Moreover, by contrast, norbixin slightly but significantly increased it (Figure 5B, *p* < 0.05). To confirm this observation, we also tested the effect of norbixin on the RAR transactivation induced by BMS 753, which is a specific RAR agonist, but we could not detect any norbixin effect (Figure 5C). To determine whether norbixin could bind RAR, we decided to perform competition experiments in vitro between norbixin and [^3^H] ATRA. We showed that norbixin bound to RAR-α but with a very low affinity, with an IC_50_ of × 94 10^−6^ M and a K_D_ of 4.7 × 10^−5^ M (Figure 5D). Altogether, the present observations suggest that the photoprotection and anti-angiogenic and anti-inflammatory effects of norbixin were not related to the inhibition of RAR transactivation but is more probably linked with the inhibition of PPAR and RXR transactivation, as reported previously [23]. In parallel, to make sure that the biological effects of BMS 195614 were specifically related to its inhibition of RAR transactivation, we checked whether BMS 195614 could also modulate the PPAR and RXR transactivations induced by A2E. Here, we showed that BMS 195614 at 10 μM inhibited the PPAR transactivation induced by A2E by 119.1% (*p* < 0.0001) (Figure 5E) and tended to decrease the level of RXR transactivation induced by A2E (Figure 5F). But this effect was not statistically significant.

### 2.6. The RAR Inhibitors AGN 193109 and BMS 493 Reduced the Phototoxicity of RPE Cells Induced by Blue Light in the Presence of A2E

Because we found that the RAR-α antagonist BMS 195614 was not specific, we aimed to select another commercially available RAR inhibitor to compare its biological activity to the effects of norbixin in our in vitro model of AMD. To do so, first, we tested the capacity of two other RAR inhibitors to protect RPE cells against A2E-induced phototoxicity in vitro. In the phototoxicity test, AGN 193109, which is a pan-RAR antagonist, at the minimal dose of 1 μM (*p* < 0.0001) (Figure 6A) and BMS 493, which is a *pan*-RAR inverse agonist, significantly restored the viability of RPE cells at the minimal dose of 0.1 μM (*p*< 0.001) (Figure 6B). The optimal effect of these RAR inhibitors was comparable with the photoprotection provided by norbixin at 20 µM (refer to Figure 2A).

### 2.7. The Commercially Available RAR Inhibitors AGN 193109 and BMS 493 Displayed Non-Specific Modulation of PPARs or RXR Transactivation in RPE Cells In Vitro

To decipher the precise roles of RAR, PPAR and/or RXR in the photoprotection of RPE cells provided by AGN 193109 and BMS 493, we tested whether these RAR inhibitors were able to reduce the transactivation of these NRs induced by A2E. Here, we showed that the pan-RAR antagonist AGN 193109 inhibited the RAR transactivation induced by A2E by 79.4% (*p* < 0.05) (Figure 7A). Moreover, AGN 193109 at 1 μM inhibited the PPAR transactivation induced by A2E by 49.9% (*p* < 0.05) (Figure 7B). By contrast AGN 193109 did not significantly inhibit the RXR transactivation induced by A2E (Figure 7C). In parallel, we report that BMS 493, which is a pan-RAR inverse agonist, strongly reduced the RAR transactivation in our cellular model by 68.9% (*p* < 0.0001) (Figure 7D). By contrast, BMS 493 at 0.1 μM did not modulate the transaction of PPAR by A2E (Figure 7E), but at the same concentration, BMS 493 partially but significantly reduced the transactivation of RXR induced by A2E (by 45.9%, *p* < 0.01, Figure 7F).

## 3. Discussion

A2E and other by-products of the visual cycle accumulate in the RPE and Bruch’s membrane of patients with AMD [6,7,8]. A2E was shown to be able to induce RPE cell toxicity [9,10,11], VEGF production [14,15] and inflammation [12,13,23]. The interaction of A2E with several NRs could explain these effects [16]. An important role for the α-isoform of the RAR in A2E induced VEGF expression and cytotoxic effects has been reported [14,15]. However, our own previous studies suggested that instead of RAR, it was PPAR and RXR transactivations that were implicated in A2E-induced inflammation and angiogenesis [23,27]. Here, using our established model of primary porcine RPE cells, we re-evaluated the roles of RAR, PPAR and RXR in A2E’s mechanisms of action.

At first, our present study led to apparently contradictory results: on one hand, we confirmed that A2E was able to induce the transactivation of endogenous RAR and the biological effects (phototoxicity, the transactivations of NF-κB and AP-1, and the expressions of IL-6 and VEGF) induced by A2E were reverted using BMS 195614, which is a selective RAR-α antagonist [24]. Moreover, other RAR inhibitors (AGN 193109 [28,29] and BMS 493 [30]) protected RPE cells against A2E stimulation in the presence of blue light (summarized in Table 1). All these observations seemed to confirm the work by Iriyama et al. [14,15]. But on the other hand, norbixin, which also protected against A2E-induced phototoxicity and was able to reduce NF-κB and AP-1 transactivations and limit the expressions of IL-6 and of VEGF [23], did not inhibit the RAR transactivation induced by A2E or BMS 753, which is a RAR-α agonist [30]. These puzzling results tend to cast doubt on the involvement of RAR in A2E’s phototoxicity and modulation of inflammation and angiogenesis.

Knowing that RO-41-5253, which is an RAR-α antagonist [21] that was used by Iriyama et al. [14,15] to propose the involvement of RAR in A2E’s biological effects, is also a PPAR-γ agonist [22], we proposed the hypothesis that perhaps the RAR inhibitors tested in the present study were not totally specific. Indeed, in agreement with this hypothesis, we demonstrated that in addition to their selective inhibition of RAR transactivation, BMS 195614 and AGN 193109 displayed nonspecific inhibitions of PPAR. In addition, although BMS 493, which is a pan-RAR inverse agonist, did not inhibit PPAR transactivation, it partially reduced the RXR transactivation induced by A2E. Therefore, we propose that the photo-protection and anti-inflammatory and anti-angiogenic effects provided by RAR inhibitors in vitro could have been due to modulations of PPAR or RXR transactivations rather than by the inhibitory properties of RAR transactivation.

Previous observations and our present results, summarized in Table 1, confirmed the importance of PPAR and/or RXR transactivation by A2E in the induction of inflammation [23,27]. In addition, PPAR and/or RXR modulation was also involved in VEGF expression and in the survival of RPE cells against blue light challenge in the presence of A2E, perhaps through a reduction in AKT phosphorylation [23].

Nevertheless, our results do not mean that RAR activation induced by A2E does not play an important role in RPE dysfunction or AMD pathophysiology. In the present study we tested only three aspects of RPE functions (cell death, inflammation and angiogenesis illustrated by VEGF expression). RAR transactivation induced by A2E could be involved in many other functions regulating retinal homeostasis, including the phagocytosis of outer segments of photoreceptors, the regulation of cholesterol and lipid metabolism [31], the epithelial–mesenchymal transition, subretinal fibrosis [32], the mitochondrial functions and the maintenance of tight junctions, among others [33]. Understanding the exact roles of RAR, PPAR and RXR in these functions could provide a better understanding of retinal homeostasis in healthy and diseased individuals.

Our results also do not preclude the importance of other NRs in explaining A2E’s biological effects by modulating several molecules that are important in RPE functions and AMD pathophysiology [20]. For instance, the Liver X receptor (LXR) was shown to modulate the expressions of APOE and ABCA1 [34]. In macrophages, LXR in conjunction with PPAR and/or RAR was also proposed to at least partially control the expression of CD36, Mer tyrosine kinase (MerTK), complement component C1q (related to complement activation) [35,36], and the expressions of VEGF [37] and MMP9 [38]. Similarly, the thyroid hormone nuclear receptor (TR), which is another NR, also deserves some attention since its inhibition was shown to protect RPE cells and photoreceptors in vitro and is neuroprotective in vivo [39,40,41]. Altogether, despite the fact that our results and the work of others point specifically on the important roles of PPAR and RXR activations in controlling several AMD-pathogenic pathways [42,43,44,45,46,47], the precise effects of many members of the NR family deserve to be studied further in order to better understand the deleterious effects of A2E on RPE cells and in the retina [20]. In addition, the present study confirmed the beneficial effect of norbixin on RPE cells in vitro and strengthened our hypothesis that norbixin and its derivative BIO203 could be developed for the treatment of early stages of AMD. Moreover, our present observation better clarifies their potential mode of action.

Of course, our study is not devoid of limitations, for instance we performed our experiments in a primary RPE cells from pig origins rather than the more broadly used ARPE-19 cell line and it would be important to reproduce our observations in this in vitro model. Nevertheless, we believe that the primary RPE cells that we used provide a more relevant model compared with ARPE-19, which although it is used in many preclinical studies, it is not completely equivalent to “natural” human RPE cells, especially in terms of phenotype and regarding their level of expression of nuclear receptors [20].

## 4. Materials and Methods

### 4.1. Reagents/Chemicals

All usual chemicals and primers were from Sigma (St. Louis, MO, USA). Reagents for the cell culture, transfection and quantitative RT-PCR were from Thermo Fisher Scientific (Waltham, MA, USA). The RNA extraction NucleoSpin^®^ RNA kit was from Macherey Nagel (Düren, Germany). The BMS 753, which an RAR-α agonist [30]; BMS 195614, which is a selective RAR-α antagonist [24]; AGN 193109, which is a high-affinity pan-RAR antagonist [28,29]; RO-41-5253, which is an RAR-selective antagonist [22]; and BMS 493, which is a pan-RAR inverse agonist [30], were purchased from TOCRIS (Bristol, UK). ECL prime and PVDF membrane were from Amersham GE Healthcare (Buckinghamshire, UK). A large dose range study from 0.1 to 20 µM to determine the optimal concentration of BMS 195614 was performed. We observed that BMS 195614 was photoprotective at 2.5 and 5 µM, but we chose to test only the 10 μM concentration in further experiments because it gave the same level of photoprotection as NBX at 20 μM, which is the molecule we developed to treat the early stages of dry AMD and was our main comparator and positive control. Similarly, dose-finding experiments for AGN 193109 and BMS493 were performed; the optimal lowest concentrations that gave RAR inhibition while avoiding any cytotoxicity in our model were 1 µM and 0.1 µM, respectively. Primary antibodies against the following proteins were used: GAPDH (Santa Cruz Biotechnology, Inc, Dallas, TX, USA) and Bcl2 (Cell Signaling, Danvers, MA, USA). Secondary antibodies were from Jackson ImmunoResearch (Cambridgeshire, UK). The Cignal Pathway Reporter Assay Kits were from QIAGEN (Frederick, MD, USA). The Dual-Luciferase Reporter Assay System was purchased from Promega (Madison, WI, USA). The pcDNA 3.1 (+)-PPAR, pcDNA 3.1 (+)-RXR and pcDNA 3.1 (+)-RAR (pig sequence) were purchased from Genscript (Piscataway, NJ, USA) or from Origene (Rockville, MD, USA).

### 4.2. Synthesis of Norbixin

9’-*cis*-norbixin was prepared from the 9’-*cis*-bixin (AICABIX P, purity 92%) purchased from Aica-Color (Cusco, Peru) by alkaline hydrolysis, as previously described [26] and according to Santos et al. [48]. Purity at 97% was determined by HPLC. The obtained product (the 9’-*cis* isomer) showed an HPLC purity of 97%, as confirmed by ^1^H nuclear magnetic resonance (using malonic acid as the internal standard). 9’-*cis*-Norbixin was stored in powder form at −80 °C until use. Fresh solutions were prepared in DMSO before each experiment.

### 4.3. Synthesis of A2E

A2E (*N*-retinylidene-*N*-retinylethanolamine) was synthesized by Orga-link (Magny-Les-Hameaux, France), as described before [26]. Briefly, all-*trans*-retinal, ethanolamine and acetic acid were mixed in absolute ethanol in darkness at room temperature over 7 days. The crude product was purified by preparative HPLC in the dark to isolate A2E with a purity of 98%, as determined by HPLC. A2E (20 mM in DMSO under argon) was stored at −20 °C. To avoid cytotoxicity to the RPE cells, we used A2E at 17.5 µM for the transactivation experiments, during which the RPE cell density (6 × 10^4^ cells/cm^2^) was lower than in the phototoxicity test and the mRNA and protein expression experiments (1.5 × 10^5^ cells/cm^2^). In these other experiments, A2E was used at 30 µM.

### 4.4. In Vitro Model of RPE Phototoxicity and Treatments

As previously described [26], pig eyes were obtained from a local slaughterhouse and transported to the laboratory in ice-cold Ringer solution. After the removal of the anterior segment of the eye, the vitreous and neural retina were separated from the RPE and removed. The eyecup was washed twice with phosphate buffer saline (PBS), filled with trypsin (0.25% in PBS) and incubated at 37 °C for 1.5 h. The RPE cells were harvested by gently pipetting, centrifuged to remove trypsin, and re-suspended in Dulbecco’s Modified Eagle Medium (DMEM) supplemented with 20% (*v*/*v*) fetal-calf serum (DMEM20%FCS) and 0.1% gentamycin. The cells were seeded into 60 mm diameter Petri dishes, cultured in an atmosphere of 5% CO_2_/95% air at 37 °C, and supplied with fresh medium after 24 h and 4 days in vitro. After one week in culture, the cells were trypsinized and transferred to a 96-well plate at a density of 1.5 × 10^5^ cells/cm^2^ in DMEM20%FCS. After 2 days in vitro, A2E was added to the medium at a final concentration of 30 μM, and 19 h later, blue light illumination was performed for 50 min using a 96 blue LED device (Durand, St Clair de la Tour, France) that emitted at 470 nm (1440 mcd, 8.6 mA). Just before illumination, the culture medium was replaced by a modified DMEM without any photosensitizer and with 2% FCS. Then, 24 h after the blue light irradiation, all cell nuclei were stained with Hoechst 33,342 and the nuclei of dead cells were stained with ethidium homodimer 2, fixed with paraformaldehyde (4% in PBS, 10 min), and 9 pictures per well were captured using a fluorescence microscope (Nikon TiE) equipped with a CoolSNAP HQ2 camera and driven by the Metamorph Premier On-Line program. The quantification of live cells was performed using Metamorph Premier Off-Line and a home-made program by the subtraction of dead cells from all cells. Cell treatments were performed as follows: All the drugs used in these experiments were prepared as stock solutions in DMSO. Drugs tested for their protective effect were added to the culture medium 48 h before illumination. For the experiments aimed at measuring the mRNA or protein expression in RPE, the cells were seeded in 24-well or 6-well plates and the treatments were performed as before, but the experiment was stopped before the illumination. For the mRNA analysis, cells were lysed using the lysis buffer from the NucleoSpin^®^ RNA kit, and the sample was stored at −80 °C. For the protein analysis, the cells were collected in Eppendorf tubes, frozen in liquid nitrogen and kept at −80 °C.

### 4.5. Studies of Binding to RAR-α

Studies of the binding of A2E and norbixin to RAR-α were performed in vitro by an external laboratory (Eurofins Cerep, Celle L’Evescault, France, https://www.eurofinsdiscovery.com/catalog/raralpha-human-retinoic-acid-nhr-binding-agonist-radioligand-assay-panlabs/269200, accessed on 31 January 2018) through competition experiments between A2E or norbixin and [^3^H]9-*cis*-retinoic acid as the natural ligand of RAR-α. Briefly, human recombinant RARα-LBD expressed in insect cells were used in a modified Tris-HCl buffer, pH 7.4. Non-specific binding was estimated in the presence of 1 μM 9-cis-retinoic acid. Receptors were counted to determine the [^3^H]9-cis-Retinoic acid that was specifically bound.

### 4.6. Protein Analysis

RPE samples were lysed in 20 mM Tris-HCl, 150 mM NaCl, 1 mM EDTA and 1% NP-40 buffer (pH 7.5) containing a cocktail of protease and phosphatase inhibitors. Equal amounts of proteins (30 μg of protein per well) were resolved by 12% SDS polyacrylamide gel electrophoresis and electro-transferred onto a PVDF membrane using a standard protocol. The membranes were blocked with 5% milk for 1 h, followed by incubation with a primary antibody overnight at 4 °C. Subsequently, the membranes were incubated with the corresponding horseradish-peroxidase-conjugated secondary antibodies for 1 h. The signal was developed using enhanced chemiluminescence (ECL) reagents prime detection kit, quantified by densitometry using Bio1D (Vilber Lourmat, Germany) and normalized by GAPDH levels. Each experiment was done at least 3 times.

### 4.7. Quantitative RT-PCR

Total RNA was extracted using the NucleoSpin^®^ RNA kit according to manufacturer’s instructions. RT of 500 ng of RNA was performed using the SuperScript III Reverse Transcriptase following the manufacturer’s instructions. A total of 5 ng of cDNA was amplified using the SYBR GREEN real-time PCR method. PCR primers for IL-6 and VEGF and the housekeeping gene GAPDH (Table 2) were designed using Primer3Plus Bioinformatics: https://www.bioinformatics.nl/cgi-bin/primer3plus/primer3plus.cgi accessed on 31 January 2018. The RT-PCR using the StepOne Plus (Life Technologies, Carlsbad, CA, USA) consisted of incubation at 50 °C for 5 min, followed by 40 cycles of 95 °C for 15 s and of 60 °C for 1 min. The reaction was completed by a melt curve stage at 95 °C for 15 s, 60 °C for 1 min, 95 °C for 15 s and a final step at 60 °C for 15 s. The relative mRNA expression was calculated using the comparative threshold method (Ct-method) with GAPDH for normalization, as previously described [27]. All experimental conditions were processed in triplicate and each experiment was done at least 3 times.

### 4.8. RAR, PPAR, RXR, AP-1 and NF-κB Transactivation Assays

After one week in culture, the cells were trypsinized and transferred to a 96-well plate at a density of 6 × 10^4^ cells/cm^2^ in DMEM20%FCS. The next day, cells were transfected using the Cignal Reporter Assay Kits for RAR, PPAR, RXR, AP-1 and NF-κB according to the manufacturer’s specifications. Transfection was performed with Lipofectamine and Plus Reagent in a serum-free medium. Three hours after the transfection, the medium was replaced and treatments with the molecules to assay were performed. In these transactivation experiments, a concentration of A2E of 17.5 µM was used to avoid cytotoxicity to the RPE cell layer seeded with a lower density than in other tests (6 × 10^4^ cells/cm^2^ vs. 1.5 × 10^5^ cells/cm^2^). The luciferase activity was measured the next day using the Dual-Luciferase Reporter Assay System. The measurements were performed with a luminometer (Infinite M1000 from Tecan, Mannedorf, Switzerland). Firefly/Renilla activity ratios were calculated for each condition and ratios from transcription-factor-responsive reporter transfections were divided by ratios from negative control transfections to obtain the relative luciferase unit, as described by the manufacturer. At least 3 independent transfections were performed in triplicate for each condition.

### 4.9. Statistical Analyses

Statistical significance was determined by applying an analysis of variance (one-way ANOVA for one parameter). After a significant ANOVA, comparisons between all groups were made with Tukey’s test, or for comparison between groups and A2E treated cells only, with Dunnett’s post-test, as specified according to the homogeneity of variance. The significant threshold was fixed at 0.05, i.e., the *p*-value had to be lower than 0.05 to be significant. Tests were performed using Prism 9.3.1 (GraphPad Software, La Jolla, CA, USA).

## 5. Conclusions

In conclusion, contrary to what was previously thought [14,15], our observations indicate that the induction of PPAR or RXR, rather than RAR transactivations induced by A2E, are central to explaining the phototoxicity, pro-inflammatory and pro-angiogenic effects of A2E in RPE cells in vitro. Obviously, our experimental model using primary porcine RPE cells in vitro did not reproduce all the aspects of the complex AMD pathophysiology. Nevertheless, our results shed new light on the potential mechanisms involved in retinal degenerative diseases. More broadly, our observations indicate that commercially available RAR inhibitors display non-specific antagonism on PPAR or RXR activation. Therefore, we warn those working in the field of NRs, and particularly RAR biology, to consider whether the biological effects they observe are solely linked to RAR inhibition or could be due to a non-specific modulation of PPAR or RXR transactivation.

## Figures and Tables

**Figure 1 ijms-25-03037-f001:**
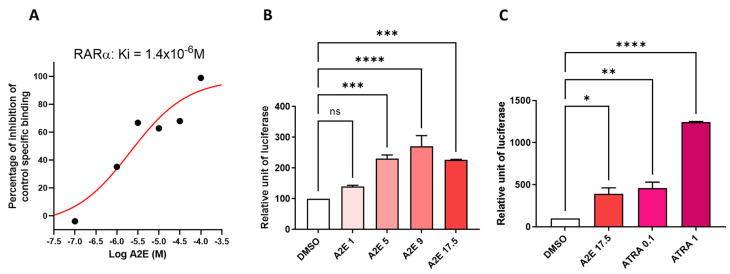
A2E bound RAR-α and induced its transactivation. (**A**) A2E binding to RAR-α. (**B**) RAR transactivation induced by increasing A2E concentrations. (**C**) Comparison of RAR transactivation induced by A2E (17.5 μM) and ATRA (0.1 and 1 μM). Bars represent mean ± SEM (n = 3–4). ns: not significant, * *p* < 0.05, ** *p* < 0.01, *** *p* < 0.001 and **** *p* < 0.0001 compared with control (DMSO) (one-way ANOVA, Tukey’s multiple comparisons tests).

**Figure 2 ijms-25-03037-f002:**
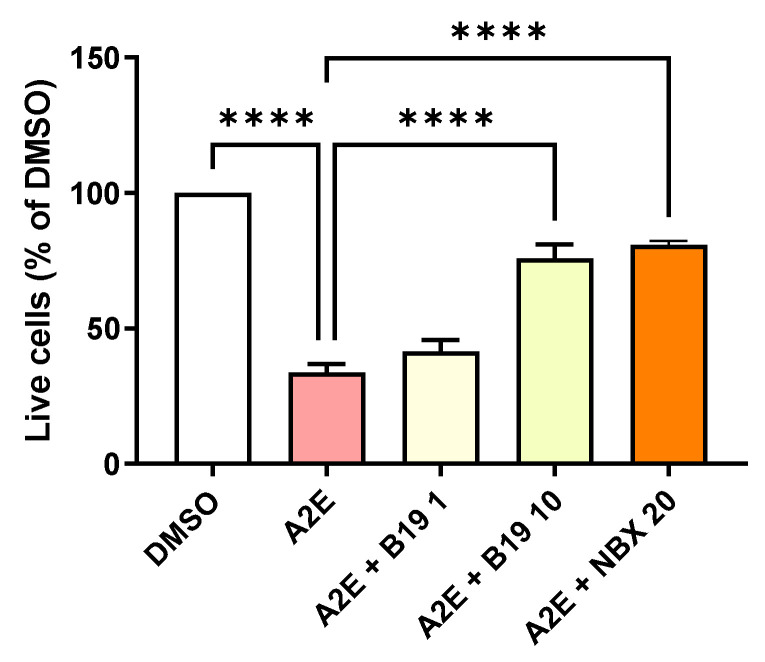
Dose response of the photoprotection induced by the RAR inhibitor BMS 195614 (B19) on RPE cells exposed to blue light in the presence of A2E. Cells treated with A2E (30 μM) and B19 at the indicated concentrations in μM. n = 3–5 per group. Photoprotection induced by NBX at 20 μM is represented in orange for comparison. Bars represent the mean ± SEM of the percentage of live cells compared with cultures of RPE cells with vehicle (DMSO). **** *p* < 0.0001 compared with cells treated with A2E alone and exposed to blue light (n = 13) (one-way ANOVA, Tukey’s multiple comparisons tests).

**Figure 3 ijms-25-03037-f003:**
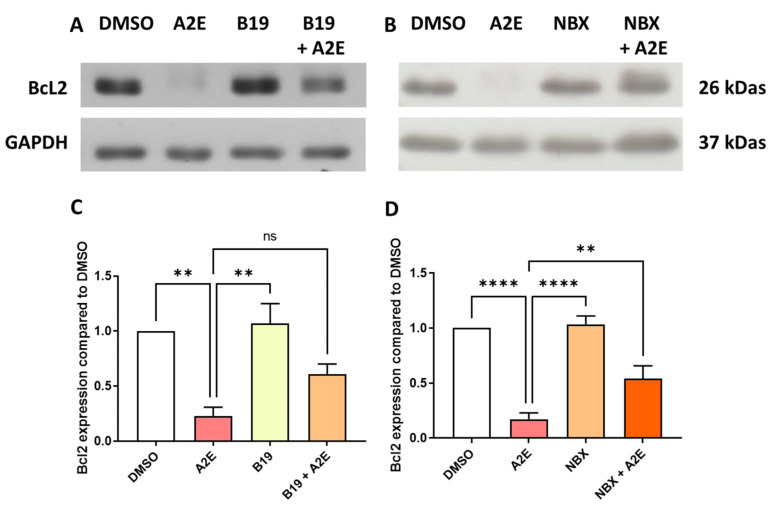
BMS 195,614 (B19) and Norbixin (NBX) restored Bcl2 protein expression downregulated by A2E. (**A**) representative western blot, and (**C**) quantification of the effects of A2E (30 μM), BMS 195614 (B19) (10 μM) and A2E (30 μM) + B19 (10 μM) on endogenous Bcl2 expression measured by Western blot (n = 3–4 per group). (**B**) representative western blot, and (**D**) quantification of the effects of A2E (30 μM), NBX (20 μM) and A2E (30 μM) + NBX (20 μM) on endogenous Bcl2 expression measured by Western blot (n = 3–4 per group). Bars represent mean ± SEM. ns: non-significant, ** *p* < 0.01 and **** *p* < 0.0001 (one-way ANOVA, Dunnett’s post-test).

**Figure 4 ijms-25-03037-f004:**
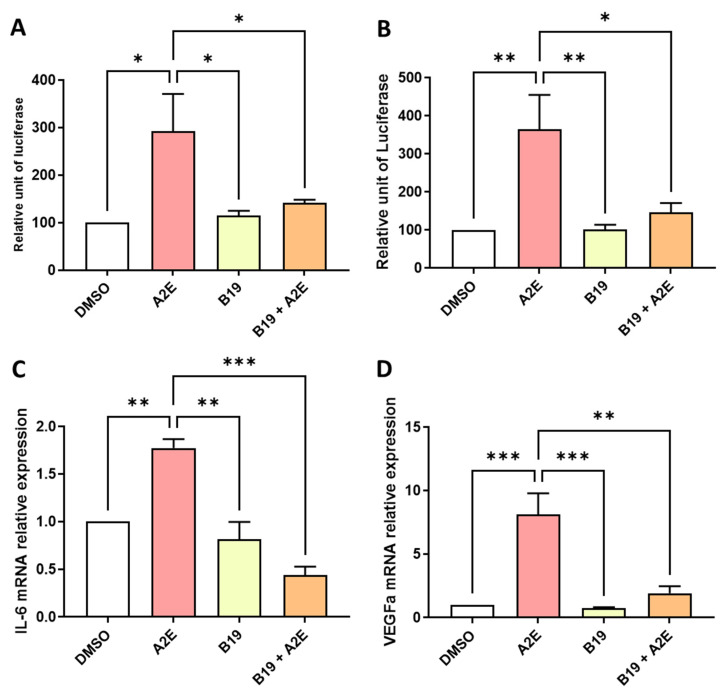
BMS 195614 (B19) inhibited the transactivation of AP-1 and downregulated the expression of IL-6 and VEGF that was increased by A2E in RPE cells in vitro. (**A**) Effects of A2E (30 µM), B19 (10 µM) alone and B19 (10 µM) + A2E (30 µM) on NF-κB and (**B**) AP-1 transactivations. (**C**) Effects of A2E (30 µM), B19 (10 µM) alone and B19 (10 µM) + A2E (30 µM) on IL-6 and (**D**) VEGF mRNA expressions. Bars represent mean ± SEM (n = 3–4). * *p* < 0.05, ** *p* < 0.01 and *** *p* < 0.001 (one-way ANOVA, Tukey’s multiple comparison test).

**Figure 5 ijms-25-03037-f005:**
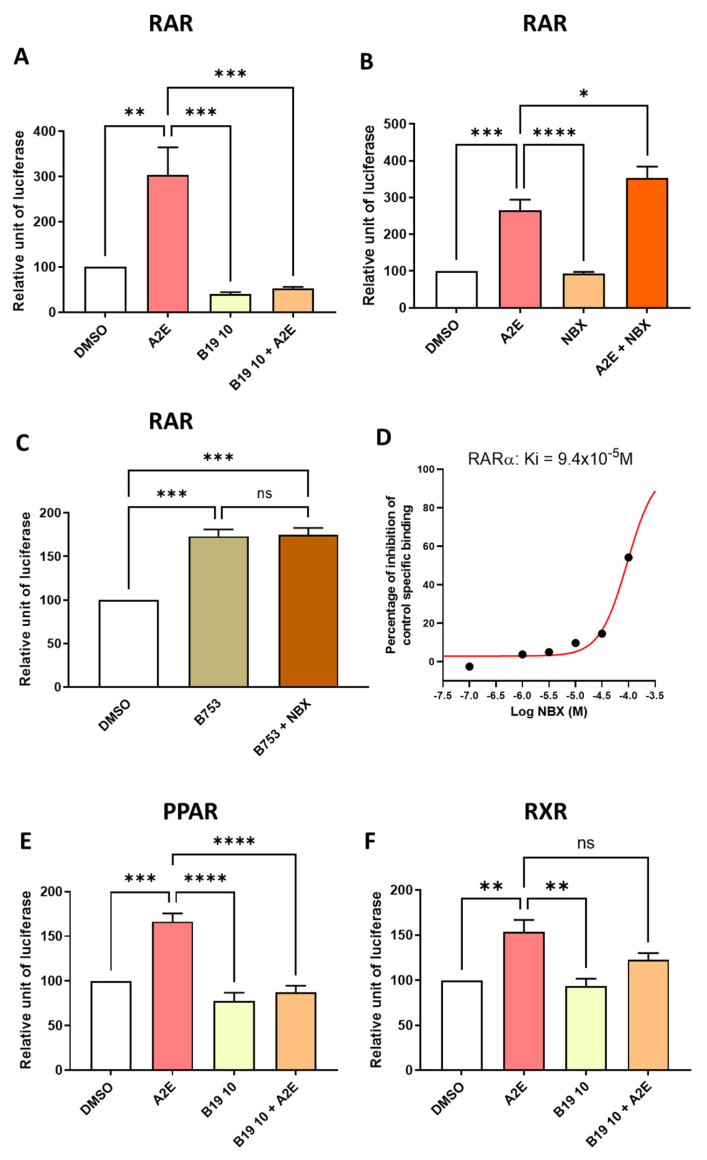
Differential effects of BMS 195614 (B19) on RAR, PPAR and RXR transactivations and effects of norbixin (NBX) on RAR transactivations induced by A2E and BMS753 in RPE cells in vitro. Effects of BMS 195614 (B19) at 10 μM (**A**) and norbixin (**B**) on endogenous RAR transactivation induced by A2E (17.5 μM). Effect of NBX (20 μM) on endogenous RAR transactivation induced by BMS 753, which is a RAR agonist (**C**). Norbixin (NBX) binding to RAR-α (**D**). Effects of BMS 195614 (B19) at 10 μM on PPAR (**E**) and RXR (**F**) transactivations induced by A2E (17.5 μM). Bars represent mean of relative luciferase activity ± SEM between cells treated with vehicle alone, B19 or NBX alone and B19 or NBX and A2E compared with cells treated with A2E or BMS753 alone. n = 3–4 per group. ns: not significant, * *p* < 0.05, ** *p* < 0.01, *** *p* < 0.001, and **** *p* < 0.0001 (one-way ANOVA, Tukey’s multiple comparison test).

**Figure 6 ijms-25-03037-f006:**
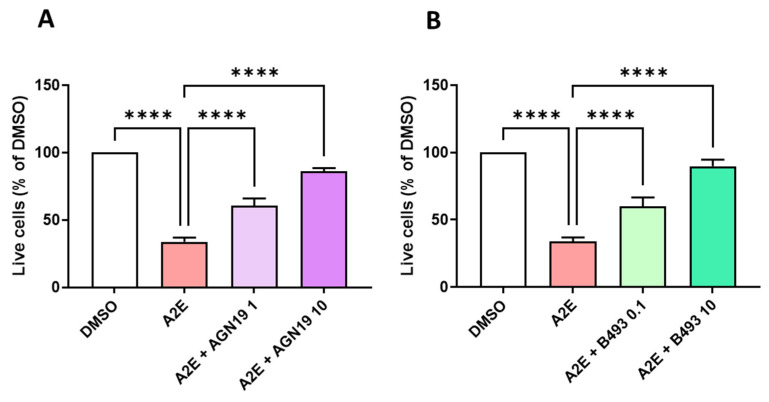
Dose response of the photoprotection induced by the RAR inhibitors AGN 193109 (AGN19) and BMS 493 (B493) on the RPE cells exposed to blue light in the presence of A2E. Cells treated with A2E (30 μM) and AGN19 (**A**) or B493 (**B**) at the indicated concentrations in μM. n = 3–5 per group. Bars represent the mean ± SEM of percentage of live cells compared with cultures of RPE cells with vehicle (DMSO). **** *p* < 0.0001 compared with cells treated with A2E alone and exposed to blue light (n = 13) (one-way ANOVA, Tukey’s multiple comparisons tests).

**Figure 7 ijms-25-03037-f007:**
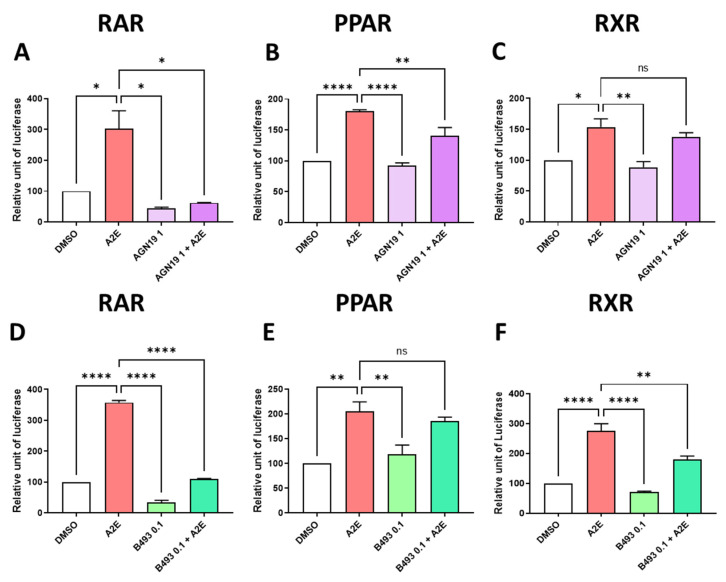
Effects of RAR inhibitors on RAR, PPAR and RXR transactivations induced by A2E in RPE cells in vitro. Effects of AGN 193109 at 1 μM AGN19, (**A**–**C**) and BMS 493 at 0.1 μM B493 (**D**–**F**) on RAR (**A**,**D**), PPAR (**B**,**E**) and RXR (**C**,**F**) transactivations induced by A2E (17.5 μM). Bars represent mean of relative luciferase activity ± SEM between cells treated with vehicle alone, RAR inhibitors alone, and RAR inhibitors and A2E compared with cells treated with A2E alone. n = 3–4 per group. ns: not significant, * *p* < 0.05, ** *p* < 0.01 and **** *p* < 0.0001 (one-way ANOVA, Tukey’s multiple comparison test).

**Table 1 ijms-25-03037-t001:** Summary of effects of RAR modulators on RPE cells.

		Effects Alone on the Transactivation of			Effects in Presence of A2E on	
Name	Nature	RAR	PPAR	RXR	RPE Survival	IL-6/VEGF
NBX	RAR-α ligand	Activation	Inhibition $	Inhibition $	Photoprotection	Inhibition
BMS 195614	RAR-α antagonist	Inhibition	Inhibition	None	Photoprotection	Inhibition
AGN 193109	Pan-RAR antagonist	Inhibition	Inhibition	None	Photoprotection	N.T.
BMS 493	Pan-RAR inverse agonist	Inhibition	None	Inhibition	Photoprotection	N.T.
RO-41-5253	RAR “specific” antagonist	Inhibition	Activation †	N.T. *	Survival in vivo ¶	Inhibition ¶

* N.T.: Not tested, references: ¶: [15], †: [22], $: [23].

**Table 2 ijms-25-03037-t002:** Probes used for mRNA quantification by RT-PCR.

Gene	Sequences	
GAPDH	F	GCTGCTTTTAACTCTGGCAA
	R	CCACAACATACGTAGCACCA
IL-6	F	CGGATGCTTCCAATCTGGGT
	R	CACAGCCTCGACATTTCCCT
VEGF A	F	GTCTGGAGTGTGTGCCCA
	R	GTGCTGTAGGAAGCTCATC

## Data Availability

The data presented in this study are available on request from the corresponding author. The data are not publicly available due to the fact they are the private property of Biophytis.

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
