# Peer review of "RAR Inhibitors Display Photo-Protective and Anti-Inflammatory Effects in A2E Stimulated RPE Cells In Vitro through Non-Specific Modulation of PPAR or RXR Transactivation"

_ijms, 2024, doi:10.3390/ijms25053037_

Round 1
Reviewer 1 Report
Comments and Suggestions for Authors
Comments on the Quality of English Language· Syntaxis of sentences must be carefully checked. For instance, the authors often do mistake in the position of comas (between subject and verb) (line 86 is an example: “ATRA, was”).
Some typos are present throughout the text.
Reviewer 2 Report
Comments and Suggestions for Authors
this article written by Fontaine et al described their use of a primary RPE model induced by A2E followed by phototoxicity to characterise the influence of selected RARa modulators. they first validated the RAR transactivation of the system, followed by cytotoxicity assessment (photoprotection), AP-1 and NFkB transactivation, IL-6 (anti-inflammatory) and VEGF expression (anti-angiogenic). However when extending their examination to PPAR and RXR transactivation they picked up the off-targets effects of the RARa modulators.
This is an important finding apparently characterisation of their pharmacology metabolism and pharmacokinetics in human become crucial.
the minor points I have are as follow:
1. fig1C the list is different from that in the legend
2. Table1 the labels are not in line with the numbers
3. was any positive control be included in the cytotoxicity measurements?
4. would be good to show representative WB on their Bcl2 measurements, and again to include a positive control.
Round 2
Reviewer 1 Report
Comments and Suggestions for Authors
The authors provided sufficient explanations to most of the requests.
Some important points are still missing, as reported below.
1) Concentration of A2E: The authors explained clearly the reasons of using different A2E concentrations. However , they should include this explanation in the material and methods as well.
2) Concentrations of RAR inhibitors: I suggest to leave 1 and 10 for all the compounds and move the other results to supplementary data.
3) The authors must provide data on Bcl2, IL-6 and VEGF also for the other RAR inhibitors or they should give a scientific rationale to be included in the manuscript to justify why they checked those parameters for the B19 only. The explanation provided by the authors is not sufficient for the final manuscript.
4) B19 concentration: The authors explained clearly the reasons of using 10 μM concentration for B19. However , they should include this explanation in the material and methods as well.
5) Original blotting images: The blottings reported in the supplementary materials are cut. The authors must provide original unmodified blottings merged with molecular weight ladder.
Comments on the Quality of English LanguageSome of the grammar mistakes and typos have not been corrected, because the authors state that they did not find the typo.
For instance: line 243 still reports "talDiscussion"
Author Response
Please see the attachement.
